# Differential Proteomics Based on TMT and PRM Reveal the Resistance Response of *Bambusa pervariabilis* × *Dendrocalamopisis grandis* Induced by AP-Toxin

**DOI:** 10.3390/metabo9080166

**Published:** 2019-08-10

**Authors:** Qianqian He, Xinmei Fang, Tianhui Zhu, Shan Han, Hanmingyue Zhu, Shujiang Li

**Affiliations:** College of Forestry, Sichuan Agricultural University, No. 211, Huimin Road in Wenjiang District, Chengdu 611130, China

**Keywords:** *Arthrinium phaeospermum* (Corda) Elli, *Bambusa pervariabilis* McClure × *Dendrocalamopsis grandis* (Q.H.Dai & X.l.Tao ex Keng f.) Ohrnb., induced resistance, parallel reaction monitoring verification, proteomics

## Abstract

*Bambusa pervariabilis* McClure × *Dendrocalamopsis grandis* (Q.H.Dai & X.l.Tao ex Keng f.) Ohrnb. blight is a widespread and dangerous forest fungus disease, and has been listed as a supplementary object of forest phytosanitary measures. In order to study the control of *B. pervariabilis* × *D. grandis* blight, this experiment was carried out. In this work, a toxin purified from the pathogen *Arthrinium phaeospermum* (Corda) Elli, which causes blight in *B. pervariabilis* × *D. grandis*, with homologous heterogeneity, was used as an inducer to increase resistance to *B. pervariabilis* × *D. grandis*. A functional analysis of the differentially expressed proteins after induction using a tandem mass tag labeling technique was combined with mass spectrometry and liquid chromatography mass spectrometry in order to effectively screen for the proteins related to the resistance of *B. pervariabilis* × *D. grandis* to blight. After peptide labeling, a total of 3320 unique peptides and 1791 quantitative proteins were obtained by liquid chromatography mass spectrometry analysis. Annotation and enrichment analysis of these peptides and proteins using the Gene ontology and Kyoto Encyclopedia of Genes and Genomes databases with bioinformatics software show that the differentially expressed protein functional annotation items are mainly concentrated on biological processes and cell components. Several pathways that are prominent in the Kyoto Encyclopedia of Genes and Genomes annotation and enrichment include metabolic pathways, the citrate cycle, and phenylpropanoid biosynthesis. In the Protein-protein interaction networks four differentially expressed proteins-sucrose synthase, adenosine triphosphate-citrate synthase beta chain protein 1, peroxidase, and phenylalanine ammonia-lyase significantly interact with multiple proteins and significantly enrich metabolic pathways. To verify the results of tandem mass tag, the candidate proteins were further verified by parallel reaction monitoring, and the results were consistent with the tandem mass tag data analysis results. It is confirmed that the data obtained by tandem mass tag technology are reliable. Therefore, the differentially expressed proteins and signaling pathways discovered here is the primary concern for subsequent disease resistance studies.

## 1. Introduction

When a plant is induced by a biologically inducing factor (pathogenic toxin, etc.), the plant will change the expression of the proteins in its cells, thereby changing the activity of related enzymes to respond to the induction. In order to understand the interactions between biologically inducible factors and host plants, we need to conduct in-depth research based on proteomics. Currently, many proteomic studies focus on the response of plants to abiotic factors, such as the effects of drought and salt content, etc. [1,2].

Proteomics is the study of all proteins expressed by a cell under specific conditions. This process includes the study of protein expression levels, post-translational modifications, and interactions between proteins and other proteins or with other biomolecules. Proteomics can thus reveal the molecular mechanisms involved in a biological process or disease at the protein level [3]. At present, isobaric tags for relative and absolute quantification (iTRAQ)/tandem mass tag (TMT) technology includes a wide range of methods for proteomics research [3,4]. The general technique utilizes four or eight isotopically encoded tags that specifically label the amino group of a peptide and then uses tandem mass spectrometry to simultaneously compare the relative or absolute content of the protein in four or eight different samples [3]. González-Fernández et al. [5] reviewed the progress of proteomics in plant fungal pathogens research, which is an excellent tool that can give us a great deal of information about fungal pathogenicity by high-throughput studies. This approach has allowed the identification of new fungal virulence factors by characterizing signal transduction or biochemical pathways and studying the fungal life cycle and their life-style. Similarly, in the plant-fungus intricacies, proteomics provides rapid insight and is expected to be one of the imminent and integrative tools in biological research [6]. Rustagi et al. [6] summarized more than 17 important plant-fungal pathogen studies that used proteomics by methods including Two-Dimensional Polyacrylamide Gel Electrophoresis (2D-PAGE), Fluorescent Two-Dimensional Difference Gel Electrophoresis (2D-DIGE), Isotope-Coded Affinity Tags (ICAT), iTRAQ, Multidimensional Protein Identification Technology (MudPIT), and Mass Spectrometry (MS).

With the developing breadth of research in plant-induced disease resistance, the idea of using a pathogen toxin as an induction factor for disease resistance has garnered more attention and recognition. Many studies around the world have incorporated this idea. Yang et al. [7] explored the toxin extracted from *Phytophthora infestans* to induce potato resistance to *Phytophthora infestans*. Hui et al. [8] used the crude toxin of leaf mold to induce the changes of defense enzymes and reactive oxygen species in tomato seedlings. Ma et al. [9] reported a toxin-induced maize resistance to *Bipolaris maydis*. They found an increase in the amount of various defense enzymes, such as peroxidase (POD) and phenylalanine ammonia-lyase (PAL) in the maize seedlings after treatment with the inducer, indicating that the plant resistance is improved. Conrath et al. [10] successfully induced partial potato resistance using the incompatible races of *Phytophthora infestans* and showed that the mechanism of action of disease resistance is that the potato produces phytoalextin. Therefore, this phytoalexin has provided the basis for the study of plant induced disease resistance. However, in the area of forest devastating disease research, there are few studies into the use of toxins as inducers for disease resistance, and the mechanism of induction, studied by using proteomics, has not been established.

*B. pervariabilis* × *D. grandis* blight is a widespread and dangerous forest fungus disease. In many provinces of China, large-scale *B. pervariabilis* × *D. grandis* death has occurred due to blight, in an area spanning more than 3000 hm^2^. This disease greatly threatens the process of returning farmland to forests and the construction of ecological barriers [11]. *B. pervariabilis* × *D. grandis* blight has been listed as a supplementary object of forest phytosanitary measures [12]. The pathogen *A. phaeospermum* belongs to Fungi, Dikarya, Ascomycota, Pezizomycotina, Sordariomycetes, Xylariomycetidae, Xylariales, Apiosporaceae, *Arthrinium* [13].

In this study, the purified toxin of *A. phaeospermum* (AP-Toxin) was obtained by Li et al. [14,15,16]. Through the results of a previous study, the best inducer was the purified pathogen toxin inactivated at 60 °C, and the concentration was 40 μg/mL [17]. The TMT technique combined with liquid chromatography mass spectrometry (LC-MS/MS) were used to explore the differential expression of proteins and metabolic pathways induced by the toxin and then inoculated by *A. phaeospermum* in *B. pervariabilis* × *D. grandis*. This work provides a theoretical basis for the in-depth study of the plant’s resistance under the action of biological inducers and lays a theoretical foundation for the management of forest diseases.

## 2. Results

### 2.1. Symptoms of B. Pervariabilis × D. Grandis after AP-Toxin Induction

*B. pervariabilis* × *D. grandis* that was induced with AP-toxin and inoculated with *A. phaeospermum* shows significant visual differences when compared with the control (sterile water + *A. phaeospermum*) (Figure 1). After treatment with AP-Toxin + *A. phaeospermum*, the leaves and branches of *B. pervariabilis* × *D. grandis* show a healthy state without symptoms of yellowing and wilting (Figure 1A). Most of the *B. pervariabilis* × *D. grandis* leaves treated with sterile water + *A. phaeospermum* have yellowing symptoms, and the branches also show signs of withering (Figure 1B). It is thus shown that the inducer can enhance the ability of the *B. pervariabilis* × *D. grandis* to resist *A. phaeospermum*.

### 2.2. Quantitative Proteomic Response to Inducer

TMT technology for protein enzymatic hydrolysis, peptide quantification and labeling, strong cation exchange (SCX) fractionation, and mass spectrometry were used to identify peptides, quantify proteins, and analyze differentially expressed proteins. According to statistical analyses of the results (Table 1), 3320 unique peptides were identified, and 1791 proteins were quantified. The percentage of total protein covered by two or more different peptides reached 45.5%, indicating that the selected TMT markers combined with mass spectrometry can effectively separate and identify proteins from the *B. pervariabilis* × *D. grandis* experimental and control groups. A differentially expressed protein was defined to be one that had a change in expression greater than 1.2-fold (up-and-down), and the significance P value levels less than 0.05 were used as a standard for screening differentially expressed proteins. The screened differentially expressed proteins were blasted by National Center for Biotechnology Information (NCBI). There were 163 differentially expressed proteins, of which 75 were up-regulated and 88 were down-regulated.

### 2.3. GO and KEGG Analysis of Differentially Expressed Proteins

The Gene Ontology (GO) functional analysis of proteins was quantified for *B. pervariabilis* × *D. grandis* considering three aspects: biological processes, molecular functions, and cell components [18] (Figure 2). The results show that a significant enrichment of biological processes mainly affects the single-organism metabolic process (GO:0044710; 80 proteins; *p* value_0.00000000141) and carbohydrate metabolic process (GO:0005975; 38 proteins; *p* value_0.000000339). They all appear to activate the up-regulation. Significant enrichment of the molecular function mainly concerns catalytic activity (GO:0003824; 101 proteins; *p* value_0.000319) and cation binding (GO:0043169; 55 proteins; *p* value_0.00000316). Finally, the affected cellular components are mainly composed of an intracellular organelle part (GO:0044446; 61 proteins; *p* value_0.0000000309) and chloroplast (GO:0009507; 59 proteins; *p* value_0.000000000000000162) (Appendix A). The bioaccumulated entries show that the multiple biological pathways of *B. pervariabilis* × *D. grandis* that were significantly changed after induction agent treatment are related to plant stress tolerance: including response to osmotic stresses, external biotic stimulus, oxidative stresses, and so on. In addition, the carbohydrate metabolic process is closely related to energy metabolism. They all appear to activate the up-regulation. Carbohydrate and energy adjustment respond to plants’ responses to pathogen stress, providing the necessary dynamic support through the up-regulation of energy metabolism-related proteins to the resistance stress pathway.

Differential proteins are classified into A–F groups, including A (Metabolism), B (Genetic information processing), C (Environmental information processing), D (Cellular processes), E (Organismal systems), and F (Human diseases). The Kyoto Encyclopedia of Genes and Genomes (KEGG) classification results indicate that the differentially expressed proteins are mainly concentrated in metabolic processes (A metabolism) (Figure 3). The metabolic pathways with the highest number of differentially expressed proteins are metabolic pathways (map01100; 53 proteins; *p* value_0.000123), biosynthesis of secondary metabolites (map01110; 39 proteins; *p* value_0.0000117), and phenylpropanoid biosynthesis (map00940; 13 proteins; *p* value_0.0000346) (Figure 3, Appendix A). The phenylpropanoid biosynthesis is related to plant defense reactions. Through a series of reactions, polyphenols such as lignin, flavonoids, and flavonols, which can be synthesized can effectively prevent the invasion of various pathogens and participate in the interaction process between plants and pathogens. Moreover, this signaling is associated with plant immune responses.

### 2.4. PPI Analysis

In the protein-protein interaction (PPI) network map (used for establishing protein networks) (Figure 4), the most significantly enriched proteins, with high connectivity, are shown in red, and the significant enrichment of the metabolic pathway is shown in dark blue. The three metabolic pathways that were significantly enriched were Biosynthesis of secondary metabolites, Phenylpropanoid biosynthesis, and Metabolic pathways. There are 16 differential proteins directly related to the Biosynthesis of secondary metabolites pathway, nine of which are up-regulated; three differential proteins are directly linked to the Phenylpropanoid biosynthesis pathway, two of which are up-regulated; 22 differential proteins are linked to it in the Metabolic pathways pathway, 11 of which are up-regulated. A total of 11 differential proteins were up-regulated in this network map, and it was found that PH01000123G1100, PH01000159G0130, PH01000713G0340, PH01000761G0570, and PH01003309G0170 were associated with plant resistance expression, both of which showed significant up-regulation and at the same time affected multiple significantly enriched metabolic pathways (Table 2).

Round nodes represent proteins/genes (for fold change analysis, red shows up-regulation and green shows down-regulation). Rectangular nodes represent KEGG pathways/biological processes.

A significant *p* value is represented by a yellow–blue gradient, where yellow indicates a small *p* value and blue indicates a large *p* value.

### 2.5. Mass Spectrometry Analysis of PRM Validation Candidate Peptides

Parallel reaction monitoring (PRM, a sort of Multiple Reaction Monitoring for peptides) mass spectrometry was used to verify the target peptides found in the TMT analysis of the AP-toxin treated *B. pervariabilis* × *D. grandis*, and the 14 candidate peptides of the target protein were subjected to LC-PRM/MS analysis. The Skyline analysis results of each candidate peptide are shown in Appendix A and include the chromatographic peak contrast map (Skyline analysis map) of each peptide segment in different samples. After the Skyline analysis of 3–5 sub-ions with high abundance and as continuous as possible in the secondary mass spectrometry of candidate peptide segments, the peak area of each target peptide segment (Appendix A) was obtained. This included the name of the target protein, sequence of peptide segments, charge number of parent ions, selected sub-ions, charge number of the sub-ions, and the original peak area of each sub-ion for quantification. The sum of all ion peak areas of each peptide segment is the ion peak area of the peptide segment. The ionic peak area of the normalized (N) corresponding peptide segment is presented in Appendix A. The normalized peak area of the peptide segment was used to quantitatively analyze the target peptide segments in different samples, and the results are shown in Appendix A.

### 2.6. PRM Verification of the Top Candidates for Differentially Expressed Proteins

A quantitative analysis using LC-PRM/MS of the expression levels of 14 candidate proteins associated with response-induced resistance was used to validate the results of TMT. The PRM quantitative results (Table 3) show similar trends to TMT for 13 candidate proteins, indicating that the data obtained by this experimental TMT technique combined with LC/MS is reliable. PH01000087G1730, PH01000860G0570, PH01000898G0600, PH01001064G0040, PH01001724G0160, PH01001918G0090, PH01002295G0170, and PH01100083G0010 exhibit PRM results close to the TMT results; however, PH01000159G0130, PH01000713G0340, PH01000761G0570, and PH01003309G0170 show multiple differences from TMT. It is possible that LC-PRM/MS quantitative analysis technology has higher sensitivity and resolution for these proteins [19].

## 3. Discussion

TMT-labeled nanoscale liquid chromatography tandem mass spectrometry (LC-MS/MS), as an emerging quantitative method technology, overcomes the shortcomings of traditional methods that cannot quantify macromolecules and proteins [20]. We used TMT technology combined with LC/MS analysis for the first time to study forest-induced resistance. The main results identified 163 differentially expressed proteins by three independent experiments, of which 75 were up-regulated and 88 were down-regulated. GO annotation and enrichment analysis of the differentially expressed proteins determined the biological process, cellular component, or molecular function of the various proteins/genes. The results indicate that the differentially expressed proteins are mainly concentrated in biological processes and cellular components. KEGG pathway annotation and enrichment analysis indicates that the differential proteins induced by AP-toxin in *B. pervariabilis* × *D. grandis* mainly enriched the metabolic pathways such as starch and sucrose metabolism, phenylpropanoid biosynthesis, the citrate cycle (TCA cycle), and amino acid transport and metabolism. The key differential proteins for screening response inducers are sucrose synthase, ATP-citrate synthase beta chain protein 1, peroxidase, and phenylalanine ammonia-lyase. By using PRM combined with LC/MS quantitative analysis to verify TMT results, 14 related proteins were screened from KEGG and GO annotation and functional classification based on the influence of plant resistance. The results showed that 13 candidate proteins showed similar trends to TMT, indicating TMT experiment. The data is reliable.

Inducing disease resistance mainly defends against disease by activating the resistance mechanism in plants and enhances preventive measures and controllability. It is an important method for current plant disease prevention and control [21,22]. Ping [23] used iTRAQ to comprehensively analyze the protein profile induced in *Arabidopsis thaliana* root, where an abundance of proteins related to the phenylpropanoid pathway, oxidative stress, and respiratory processes were highly increased. Jing et al. [24] studied induced sweet orange and found that the proteins that were up-regulated were related to the defense response. Lippert et al. [25] studied the protein profile of *Norwegian spruce* under chitosan induction and found proteins associated with oxidative stress response. Stresses in plants after induction include significant changes in various protein levels, such as pathogenesis-related proteins, protease inhibitors, polyphenol oxidase, peroxidase, lipoxygenase, and phenylalanine ammonia-lyase [26,27,28,29]. Furthermore, differentially expressed proteins such as sucrose synthase in the energy-related glucose metabolism and citrate synthase in the citric acid cycle are also significantly upregulated [30,31]. Similar to the above results, in this research, differentially expressed proteins involved in carbohydrate and energy formation processes were up-regulated, and differential proteins involved in signal sensing and amino acid biosynthesis were also significantly expressed. The adjustment of carbohydrates and energy is associated with plant response to pathogen stress [32,33]. Signal-aware metabolic pathways respond to plant resistance to pathogens [34]. The synthesis of amino acids is also considered to be an important mechanism of plant stress response and regulation [35,36]. In summary, *B. pervariabilis* × *D. grandis* was induced to activate the defense and energy metabolism-related pathways, and the necessary protein support was provided by the up-regulation of energy metabolism-related proteins. Therefore, the major differential proteins involved in these metabolic pathways are considered to be key proteins in response to inducer action.

Among the starch and sucrose metabolism and citrate cycle (TCA cycle) pathways, the differentially expressed proteins are sucrose synthase and ATP-citrate synthase beta chain protein 1. Sucrose synthase can catalyze the conversion of sucrose to uridine diphosphate glucose (UDPG) and fructose [37], a widely occurring UDP-glycosyltransferase and a key enzyme that catalyzes sucrose metabolism [38,39]. Glycogen is an important energy storage material for synthetic bio-organisms [40]. It is involved in the biosynthesis of tannins, cellulose, starch, glycolipids and glycoproteins [37], providing the energy and carbon skeleton needed for plant growth and regulating the gene expression of the signaling molecule [38]. Thus, glycogen participates in the response of plant stress against the infection of pathogenic bacteria [41]. The TCA cycle is linked to the glycolysis pathway and participates in the carbon metabolism pathway, which plays a vital role in the energy metabolism of organisms. Citrate synthase, which is differentially expressed in the TCA cycle pathway, is a key enzyme in the sugar metabolism pathway and has a high specificity. It specifically catalyzes the condensation reaction of acetyl-CoA with oxaloacetate to form citric acid and coenzyme. Citrate synthase from *Arabidopsis* mitochondria [42], releasing more energy, could accelerate physiological metabolic reactions against adverse environments. This conclusion is the same as in our study. *B. pervariabilis* × *D. grandis* significantly increases the expressions of sucrose synthase and ATP-citrate synthase beta chain protein 1 after the inducer is introduced. The reason for this may be that in the starch and sucrose metabolism and TCA cycle pathways, *B. pervariabilis* × *D. grandis* uses sugar and other substances to oxidize and obtain energy. The various metabolites produced by the cyclic process are the raw materials for the biosynthesis of many important substances in the organism, which may be related to the plant immune response [43].

In the process of resisting stress, plants can regulate the activity of various antioxidant enzymes and reduce the damage of reactive oxygen species by regulating the changes of peroxidase activity in the metabolic pathway of phenylpropanoid biosynthesis [44]. Sugarcane enhances its ability to respond to low temperatures by increasing the activity of antioxidant enzymes in the body [45]. Studies on the cold resistance of rapeseed found that peroxidase isoenzyme activity was strong and stable in cold-resistant varieties [46]. The above conclusions are similar to the conclusions of this experiment. After the bamboo was induced by AP-toxin, the activity of peroxidase significantly changed. Perxidase (POX) is related to respiration, photosynthesis, and the oxidation of auxin. It removes excess reactive oxygen species from *B. pervariabilis* × *D. grandis*, maintains the active oxygen balance of the *B. pervariabilis* × *D. grandis*, and avoids damage to the *B. pervariabilis* × *D. grandis* cells by O_2_^−^ [47].

The differentially enriched protein in the amino acid transport and metabolism pathway is phenylalanine ammonia-lyase. Phenylalanine ammonia-lyase catalyzes the deamination of phenylalanine to trans-cinnamic acid in the phenylpropane metabolic pathway, and then synthesizes coumaroyl-CoA through a series of catalytic reactions to finally synthesize polyphenolic compounds such as lignin, flavonoids, and flavonols [48]. PAL is a key enzyme in the specific synthesis pathway of lignin monomers and plays a crucial role in lignin biosynthesis [49]. Lignin [50] is not only the basic component of cell walls, but also participates in the process of cell wall lignification. The natural barrier formed by lignin and cellulose can effectively prevent the invasion of various pathogens [51] and participate in the signal of plant-pathogen interactions. The conduction process [52] is associated with plant immune responses [53]. Flavonoids and flavonols have also been considered as screening indicators for host disease resistance in previous studies [54]. Cinnamic acid can also form benzoic acid by oxidation and then hydroxylate to form salicylic acid. Studies have shown that PAL, as a key enzyme and rate-limiting enzyme in the phenylpropane synthesis pathway, is a branching point that controls primary metabolism to secondary metabolism and can respond to a variety of biotic and abiotic stresses, such as pathogen infection and biological factor induction [55]. In this study, phenylalanine ammonia-lyase was significantly up-regulated after the action of the *B. pervariabilis* × *D. grandis* inducer, which was consistent with previous studies.

Although the study found many important related proteins for *B. pervariabilis* × *D. grandis* response to AP-toxin, there is still a lack of understanding regarding its intrinsic mechanism of action. The function of the corresponding proteins and metabolic pathways can only be clarified by further combining physiological and biochemical research and verification of gene function. This would provide a clearer understanding of the mechanism of action for the corresponding proteins. Therefore, future work should use targeted proteomics technology to mine specific anti-reverse related proteins, verify their function and structure, and clone the corresponding genes. Such studies could use transgenic means to elucidate the expression regulation mechanism of the related genes/proteins and reveal the molecular regulatory network of *B. pervariabilis* × *D. grandis* to AP-toxin induction. Finally, such work would provide technical guidance for the breeding and genetic improvement of blight-resistant *B. pervariabilis* × *D. grandis* varieties, as well as better applications in production practice.

## 4. Materials and Methods

### 4.1. Materials

Plants: one-year-old *B. pervariabilis × D. grandis* planted in the cultivation area of Sichuan Agricultural University (103°01′N, 29°54′E, elevation 515.9 m, annual average temperature 16 °C, average annual precipitation 1400–1700 mm) in Sichuan, China.

Strain: *Arthrinium phaeospermum* (Corda) Ellis, isolated from *B. pervariabilis × D. grandis* blight, was provided by the Forest Protection Laboratory of Sichuan Agricultural University of Sichuan Province, China [11]. The pathogen suspension stored at −20 °C was activated on a PDA plate and cultured at 25 °C for 5 days. Dilute pathogen spores were added at a concentration of 10^−7^ to sterile water.

Purified AP-toxin was prepared by the Forest Conservation Laboratory of Sichuan Agricultural University [14,15,16] and configured as the inducer (diluted to 40 μg/mL with sterile distilled water and inactivated at 60 °C) [17].

### 4.2. Plant Sample Processing

Thirty plants of one-year-old *B. pervariabilis* × *D. grandis* with uniform growth were selected, and the pH of the induction was adjusted to 6–8. Fifteen plants were randomly selected and sprayed on the upper eight shoots by the inducer (inactivated AP-toxin) in 4.1. The remaining 15 plants were treated with sterile water, subjected to bagging moisturizing, and sprayed once every 12 h, 3 times in total. After three days, all of the *B. pervariabilis* × *D. grandis* plants underwent acupuncture inoculation with the *A. phaeospermum* suspension (the acupuncture site was at the tip of the twig, not pierced). Dropping the *A. phaeospermum* suspension at the wound was suitable to fill the shoots without flow, at about 0.05 mL. The plant underwent further bagging and moisturizing; and samples were collected after about 30–40 days. The samples were quickly stored in liquid nitrogen for future use.

### 4.3. Protein Extraction, Enzymatic Hydrolysis and Peptide Quantification

We took a 300 μg aliquot of each sample, froze them with liquid nitrogen, and ground them into powder. We took an appropriate amount of powder from each sample, added 200 μL of lysate, sonicated them, and subjected them to TCA-acetone precipitation at −20 °C overnight. A 16,000 *g* centrifugation was conducted for fifteen minutes at −4 °C, the precipitate was washed twice with cold acetone and dried in air. Then, 150 μL of lysate was added to each tube sample, sonicated, centrifuged at 16,000 *g* for 15 min, and the supernatant was taken. Protein quantification was performed using the bicinchoninic acid (BCA) method (Beyotime Institute of Biotechnology, Shanghai, China) [56].

We took 300 μg aliquot of each sample for enzymatic hydrolysis [57], added DL-Dithiothreitol (DTT) to 100 mM of the sample, boiled in water for 5 min, and cooled to room temperature. We added 200 μL of Urea (UA) buffer (8 M urea, 150 mM Tris-HCl, pH 8.0), mixed, transferred it to a 10 KD ultrafiltration centrifuge tube, and centrifuged the sample at 12,000 *g* for 15 min. We added 200 μL of UA buffer and centrifuged it at 12,000 *g* for 15 min, then discarded the filtrate. We added 100 μL of IAA (50 mM IAA in UA), shook it at 600 rpm for 1 min, avoiding light, left it at room temperature for 30 min, and centrifuged at 12,000 *g* for 10 min. We added 100 μL of UA buffer, centrifuged at 12,000 *g* for 10 min, repeating these two times. We added 100 μL of NH_4_HCO_3_ buffer (Sigma-Aldrich, St. Louis, USA), centrifuged at 14,000 *g* for 10 min, repeating these two times. We added 40 μL trypsin buffer (6 μg Trypsin in 40 μL NH_4_HCO_3_ buffer), shook it at 600 rpm for 1 min, and then kept it at 37 °C for 16–18 h. We replaced the collection tube, centrifuged at 12,000 *g* for 10 min, collected the filtrate, added an appropriate amount of 0.1% trifluoroacetic acid solution, desalinated the C18 Cartridge, quantified the peptide, and determined the peptide concentration by OD280.

### 4.4. TMT Peptide Labeling and Peptide Grading

One hundred milligrams of peptide were taken from each sample and labeled according to the Thermo Fisher TMT Labeling Kit instructions. The experimental induction group and the control group were labeled as TMT-126 and TMT-127, and three replicates were set. The labeled peptides were mixed in equal amounts, and the dried peptides were fractionated using High-pH (Pierce™ High pH Reversed-Phase Peptide Fractionation Kit, Thermo Fisher Scientific, Waltham, USA). The sample collection was eventually combined into ten components. The peptides of each component were dried and reconstituted with 0.1% FA for LC-MS analysis. TMT markers and LC-MS data acquisition were determined by Shanghai Baipu Biotechnology Co., Ltd. (Shanghai, China).

### 4.5. Liquid Chromatography Mass Spectrometry (LC-MS/MS) Analysis

The reconstituted peptide solution was subjected to LC-MS/MS analysis, and each fraction was injected once for a total of 10 mass spectrometry analyses. Separation was performed using a nanoscale flow rate HPLC liquid phase system Easy-nLC1200 (Thermo Fisher Scientific, Waltham, USA). Buffer: solution A was 0.1% formic acid in water, and solution B was 0.1% formic acid in acetonitrile (98% acetonitrile). The column was equilibrated with 95% A solution. The sample was loaded onto a Trap column (2 cm × 100 μm, 5 μm: C18) and separated by an analytical Thermo scientific EASY column (75 μm × 120 mm, 3 μm: C18) at a flow rate of 300 nL/min. The relevant liquid phase gradient was as follows: 0–2 min, with a linear gradient of B liquid from 4–7%; 2–57 min, with a linear gradient of B liquid from 7–30%; 57–62 min, with a linear gradient of B liquid from 30–45%; 62–67 min, with a linear gradient of B liquid from 45–90%; 67–75 min, with B liquid maintained at 90%. The peptides were chromatographed and analyzed by tandem mass spectrometry using a Q-Exactive Plus Mass Spectrometer (Thermo Scientific). Analysis duration: 75 min. Detection method: positive ion. Parent ion scanning range: 300–1800 m/z. The mass-to-charge ratio of peptide and polypeptide fragments was collected as follows: 20 fragment maps (MS2 scan, higher-energy collisional dissociation (HCD)) were acquired after each full scan. Primary mass spectrometry resolution: 70,000 @ m/z 200, AGC target: 1e6, Level 1 Maximum IT: 50 ms. Secondary MS resolution: 17,500 @ m/z 200. AGC target: 1e5. Level 2 Maximum IT: 50 ms, MS2 Activation Type: HCD. Isolation window: 1.6 Th. Normalized collision energy: 35.

### 4.6. Mass Spectrometry Data Analysis

The mass spectrometry proteomics data have been deposited to the ProteomeXchange Consortium (http://proteomecentral.proteomexchange.org) via the iProX partner repository [58] with the dataset identifier PXD014363, the subject ID is IPX0001646002.

The original RAW file obtained by LC-MS/MS was imported into MaxQuant software (version 1.0.6.16) for database searching. In this experiment, the protein database translated by the *B. pervariabilis* × *D. grandis* transcriptome (All-gene.fa) file was selected for retrieval and analysis. The main library parameters were set as in Table 4.

### 4.7. Bioinformatics Analysis

Bioinformatics analysis of the *B. pervariabilis* × *D. grandis* induction group and control group was performed. The functional annotation of proteins was performed using the Blast2 GO database program (http://geneontology.org/), while the KEGG database (http://www.genome.jp/kegg/) was used to classify and group identified proteins. The Fisher’s exact test was used to analyze the significance level of each pathway and certain GO term protein enrichments, and the target proteins were analyzed by the GO [18] and KEGG [59] pathways. We searched for the protein genes in the IntAct molecular interaction database to obtain the mutual interaction data between proteins (PPI). The results were downloaded and imported into the Cytoscape package (version 3.2.1) in extensible graph markup and modelling language (XGMML) format to visualize and further analyze the function of the differentially expressed proteins. In addition, the importance of the proteins in the PPI network was assessed by calculating the degree of interaction for each protein.

### 4.8. PRM Verification

In order to verify the accuracy of TMT and LC-MS/MS data analysis, important target proteins were selected for targeted shotgun mass spectrometry, in order to directly compare the abundance of the differentially expressed proteins. We imported the PRM analysis protein information into the software Xcalibur for PRM method settings. The PRM mass spectrometry RAW file of the hybrid bamboo induction group and the experimental group was opened using the software Xcalibur 4.0.

We took 2 μg of peptide from each sample for LC-PRM/MS analysis, and performed chromatographic separation using a nanoliter flow rate Easy nLC 1200 chromatography system (Thermo Scientific) after loading. Buffer: solution A was 0.1% aqueous formic acid, and solution B was 0.1% formic acid, acetonitrile, and water (95% acetonitrile). The column was equilibrated with 95% solution A. The sample was injected into a trap column (100 μm × 20 mm, 5 μm, C18, Dr. Maisch GmbH, Ammerbuch-Entringen, Germany) and subjected to gradient separation through a chromatography column (75 μm × 150 mm, 3 μm, C18, Dr. Maisch GmbH) at a flow rate of 300 nL/min. The liquid phase separation gradient was as follows: 0–5 min, with a linear gradient of liquid B from 2% to 5%; 5–45 min, with a linear gradient of liquid B from 5% to 23%; 45–50 min, with a linear gradient of liquid B from 23% to 40%; 50–52 min, with a linear gradient of liquid B from 40% to 100%; 52–60 min, with liquid B maintained at 100%. Peptide fragmentation and targeted PRM mass spectrometry were performed using a Q-Exactive Plus Mass Spectrometer (Thermo Scientific). The analysis time was 75 min. Detection mode: positive ion. Parent ion scanning range: 350–1500 m/z. Primary mass spectrometry resolution: 70,000 @ m/z 200. AGC target: 3e6. First-order mass spectrometer maximum IT: 200 ms. Peptide secondary mass spectrometry was performed as follows: after each full scan (Full MS scan), the precursor m/z of 18 target peptides was sequentially selected according to the inclusion list for secondary mass spectrometry analysis (MS2), in which MS2 was resolved. The rate was 17,500 @ m/z 200. AGC target: 3e6, secondary mass spectrometry maximum IT: 100 ms. MS2 activation type: HCD. Isolation window: 2.0 Th. Normalized collision energy: 27. The mass spectrometry proteomics data have been deposited to the ProteomeXchange Consortium (http://proteomecentral.proteomexchange.org) via the iProX partner repository [58] with the dataset identifier PXD014364, the subject ID is IPX0001651001. The resulting mass spectrum RAW file was analyzed using the software Skyline 4.1 [60] for PRM data.

### 4.9. Data Analysis Software

The data of the experimental induction group and the control group were collected three times, and the data were statistically analyzed using SPSS v20.0. The significance of differences was determined by Student’s *t* test, and the significance level was *p* < 0.05. All data shown are the means ± standard deviations (SDs) (*n* = 3).

## Figures and Tables

**Figure 1 metabolites-09-00166-f001:**
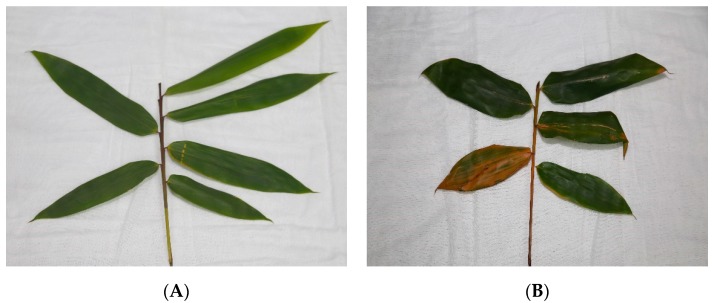
Photos of characteristic samples of *B. pervariabilis* × *D. grandis* after *A. phaeospermum* (AP) infection. (**A**) Treated with AP-toxin, and (**B**) treated with water before infection with *A. phaeospermum.*

**Figure 2 metabolites-09-00166-f002:**
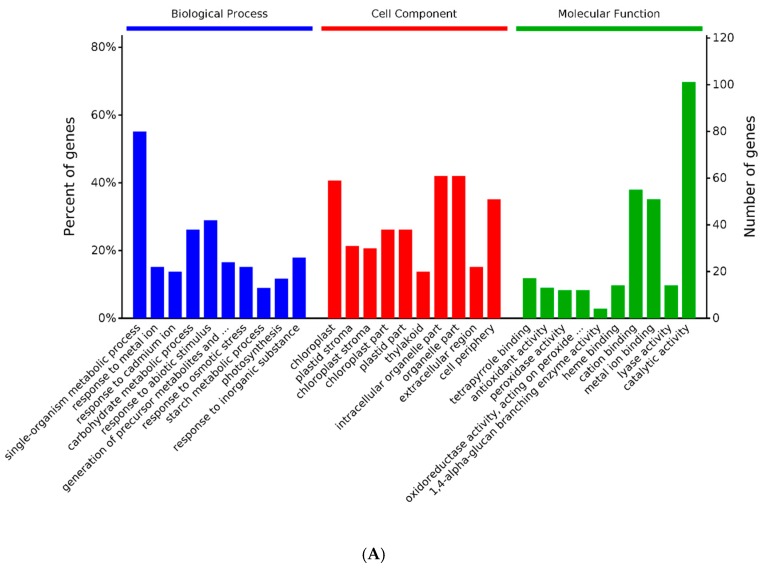
Gene ontology (GO) analysis results for significant protein enrichment between the *B. pervariabilis* × *D. grandis* induced by AP-toxin and the control group. (**A**) The top 10 most significant enriched entries in the three categories biological processes (BP), cell component (CC), and molecular function (MF). The *p* value is set to 0.05, and the items in the same category are sorted by the *p* value. The left *y*-axis represents the percentage, and the right *y*-axis represents the number of genes/proteins enriched under a certain category. (**B**) The results for the biological processes that are significantly enriched, (**C**) the results for the significantly enriched cellular components, and (**D**) the results for the molecular functions that are significantly enriched. In (**B**,**D**), the horizontal axis indicates the percentage of genes/proteins enriched in each case, and each bar is followed by the number of proteins and the corresponding *p* value.

**Figure 3 metabolites-09-00166-f003:**
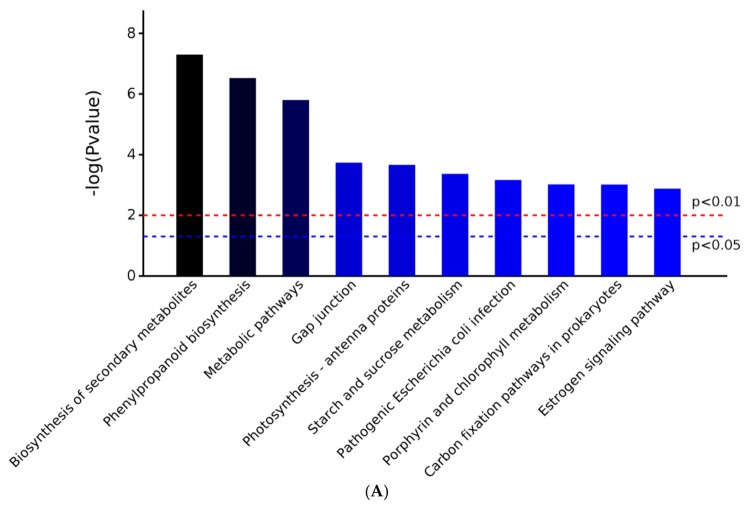
Kyoto Encyclopedia of Genes and Genomes (KEGG) pathways with significant protein enrichment between the *B. pervariabilis* × *D. grandis* induced by AP-toxin and the control group. (**A**) The top 10 KEGG signaling pathways with the most significant *p* values. Two limits of *p* values, 0.05 and 0.01, are indicated. (**B**) KEGG pathway classification indicates the signal pathways that are significantly enriched. Each bar is followed by the number of proteins and the corresponding *p* value. The KEGG pathway is divided into the following categories: A: metabolism, B: genetic information processing, C: environmental information processing, D: cellular processes, E: organismal systems, F: human diseases.

**Figure 4 metabolites-09-00166-f004:**
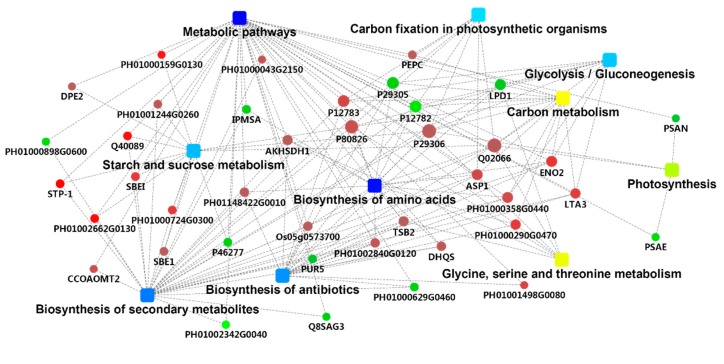
Protein-protein interaction (PPI) network map of the differentially expressed proteins in the *B. pervariabilis* × *D. grandis* between the induced group and the control group.

**Table 1 metabolites-09-00166-t001:** Statistics on the protein identification results.

Identification Result	Unique Peptides	Quantified Proteins	Up-Regulated	Down-Regulated	Significantly Different Proteins
Total	3320	1791	75	88	163

**Table 2 metabolites-09-00166-t002:** Differentially expressed protein information for high connectivity in the PPI analysis.

Protein ID	Description	KEGG Pathways
PH01000123G1100	porphobilinogen deaminase	Biosynthesis of secondary metabolites
		Metabolic pathways
PH01000159G0130	sucrose synthase	Starch and sucrose metabolism
PH01000713G0340	Adenosine triphosphate (ATP)-citrate synthase beta chain protein 1	Citrate cycle (Tricarboxylic acid (TCA) cycle)
		Energy production and conversion
PH01000761G0570	Peroxidase	Phenylpropanoid biosynthesis
		Carbohydrate transport and metabolism
PH01003309G0170	Phenylalanine ammonia-lyase	Phenylalanine metabolism
		Phenylpropanoid biosynthesis
		Secondary metabolites biosynthesis, transport and catabolism

**Table 3 metabolites-09-00166-t003:** Quantitative results for 14 candidate proteins determined using the PRM and tandem mass tag (TMT) methods.

Protein Name	Peptide Sequence	AP-Toxin + *A. phaeospermum*	Sterile Water + *A. phaeospermum*	PRM Fold Change	TMT Fold Change	Consistency between PRM and TMT
PH01000087G1730	AVAHQPVSVAIEAGGR	1,756,917.422	1,409,646.38	1.25	1.52	Yes
PH01000123G1100	TTGDMILDKPLADIGGK	3,128,973.576	3,315,437.047	0.94	1.62	No
PH01000133G0340	LVENDEVVR	6,132,972.898	4710,029.37	1.30	2.01	Yes
PH01000159G0130	QQGLNITPR	30,652,687.98	5,578,763.154	5.49	3.27	Yes
PH01000713G0340	FGGAIDDAAR	21,475,599.31	8,257,753.841	2.60	1.63	Yes
PH01000761G0570	NNPSDIDPSLNPSYAK	4,349,530.158	1,633,319.523	2.66	1.59	Yes
PH01000860G0570	DVDLSTYK	2,738,630.087	1,653,255.404	1.66	1.54	Yes
PH01000898G0600	MGNINPLTGTAGQIR	66,407,505.35	444,831,682.9	0.15	0.35	Yes
PH01001064G0040	EHLIAGWAPK	7,994,610.908	4,179,165.695	1.91	1.44	Yes
PH01001724G0160	VNVYYNEASCGR	4,996,071.907	3,097,515.346	1.61	1.57	Yes
PH01001918G0090	APDFEAEAVFDQEFIK	1,768,290.264	3,727,484.2	0.47	0.64	Yes
PH01002295G0170	VVVSSCGHDGPFGATGVK	11,037,762.06	6,282,695.56	1.76	1.45	Yes
PH01003309G0170	VGQVAAVAQAK	63,790,370.98	19,648,147.32	3.25	2.24	Yes
PH01100083G0010	YFSAAASQALDTAER	2,771,584.723	16,486,693.42	0.17	0.46	Yes

**Table 4 metabolites-09-00166-t004:** MaxQuant search library parameter settings.

Item	Value
Type	Reporter ion MS2
Isobaric labels	TMT 6plex
Enzyme	Trypsin
Reporter mass tolerance	0.005 Da
Max missed cleavages	2
Main search peptide tolerance	4.5 ppm
First search peptide tolerance	20 ppm
Mass spectrometry (MS) /MS tolerance	20 ppm
Fixed modifications	Carbamidomethyl (C)
Variable modifications	Oxidation (M), Acetyl (Protein N-term)
Database	Bamboo.fasta
Database pattern	Target-Reverse
Peptide spectrum matches false discovery rate (PSM FDR)	≤0.01
Protein FDR	≤0.01
Protein quantification	Razor and unique peptides were used for protein quantification.

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
