# Peer review of "Differential Proteomics Based on TMT and PRM Reveal the Resistance Response of Bambusa pervariabilis × Dendrocalamopisis grandis Induced by AP-Toxin"

_metabolites, 2019, doi:10.3390/metabo9080166_

Round 1

Reviewer 1 Report

This is an interesting study using TMT labelling to identify differentially expressed proteins involved in resistance response in fungal forest disease.

It is recommended the authors remove some parts of Figures 2, 3 and all of Figure 5 to supplementary material, as they do not add much to the actual body of the manuscript.

A few minor corrections;

P4, lines 132-134 – some spaces needed before terms in brackets.

P13, line 338 – change ‘We took and 300 µg of each sample ..’ to ‘We took a 300 µg aliquot of each sample …

Author Response

Point 1: It is recommended the authors remove some parts of Figures 2, 3 and all of Figure 5 to supplementary material, as they do not add much to the actual body of the manuscript.

Response 1: We appreciate the reviewer’s important comment. We think Figures 2 and 3 are very important in the result. In Figure 2, (A) indicates the top ten most significant enriched entries in the three categories biological processes (BP), cell component (CC), and molecular function (MF). (B) indicates the results for the biological processes that are significantly enriched, (C) indicates the results for the significantly enriched cellular components, and (D) indicates the results for the molecular functions that are significantly enriched. The four figures are the main results in Gene ontology (GO) analysis. Similarly, in Figure 3, (A) and (B) are the main results in Kyoto Encyclopedia of Genes and Genomes (KEGG) pathways.

     Because the results of the Figures 2 and 3 did not get too much explanation in the original manuscript, we have added the explanation of Figures 2 and 3 in our revised manuscript, and hope the highlight the importance of these figures. These added contents are as below:

Part of Figure 2: The results show that a significant enrichment of biological processes mainly affects the single-organism metabolic process (GO:0044710; 80 proteins; Pvalue_0.00000000141) and carbohydrate metabolic process (GO:0005975; 38 proteins; Pvalue_0.000000339). They all appear to activate the up-regulation. Carbohydrate and energy adjustment respond to plants’ responses to pathogen stress, providing the necessary dynamic support through the up-regulation of energy metabolism-related proteins to the resistance stress pathway.

Part of Figure 3: The metabolic pathways with the highest number of differentially expressed proteins are metabolic pathways (map01100; 53 proteins; Pvalue_0.000123), biosynthesis of secondary metabolites (map01110; 39 proteins; Pvalue_0.0000117), and phenylpropanoid biosynthesis (map00940; 13 proteins; Pvalue_0.0000346). The phenylpropanoid biosynthesis is related to plant defense reactions. Through a series of reactions, polyphenols such as lignin, flavonoids and flavonols, which can be synthesized can effectively prevent the invasion of various pathogens, and participate in the interaction process between plants and pathogens. Moreover, this signaling is associated with plant immune responses.

In addition, according to your comment, Figure 5 has been removed to supplementary material as Figure S1.

Point 2: P4, lines 132-134 – some spaces needed before terms in brackets.

Response 2: We are very sorry for our negligence. The spaces have been added in lines 132-134 in the revised manuscript.

Point 3: P13, line 338 – change ‘We took and 300 µg of each sample ..’ to ‘We took a 300 µg aliquot of each sample …

Response 3: We have made correction according to your comment. This sentence has been changed with “We took a 300 μg aliquot of each sample, froze them with liquid nitrogen, and……”.

Special thanks to you for your good comments.

Reviewer 2 Report

The authors are continuing the research of B. pervariabilis × D. grandis blight, since they have published previously in this field. In this paper they intended to carry on a differential quantitative proteomics study to find out what are the proteins up-down regulated upon induction of the infection symptoms using a purified and inactivated fungal toxin from Arthrinium phaeospermum.

The introduction is well written even if the phenomenon of immunity induction by inactivated elicitors from fungi is not well documented by other examples/references from other species.

Results showed a gradual explanation of the techniques used and the graphs are quite illustrative of the control versus elicited response treatment. Not so evident results the reading of the Figure 2 and 3 which does not get too much explanation either in the results or in the discussion places, leaving the reader to a more visual understanding of what is going on without focusing on the key elements being up/down regulated. The Supplementary figure 1 (S1) is showing the Gene Ontology accession up/down regulation sorted according P values but no info on   other column legend meaning (i.e. PAS_Value PAS_Zscore max_zscore min_zscore, etc.).

Furthermore a lot of abbreviation are not given as meaning (not even an abbreviation section exists): PPI = protein-protein interaction (used for establishing protein networks). Section 2.4 describe PPI analysis but leaves a lot of results from the Figure 4 without further interpretation leaving only the primary and secondary metabolite analysed in the discussion, meanwhile the antibiotics node-cluster is forgotten... 

Other abbreviations to explain for the reader: PRM, Parallel reaction monitoring ( a sort of Multiple Reaction Monitoring for peptides), etc.

At the end the research output is restricted to 13 top differentially expressed protein candidates (Table 3) that discussed further for their possible involvement in the disease progression after exposure to the elicitor.

Still is very poor understood the response to AP-toxin, regarding its intrinsic mechanism of action, but now after this study maybe proteins involved in plant immunity could be sorted out from the KEGG databases sorted table (S2) and put into a contest, trying to understand their possible involvement.

Material and Methods seems to be well organized even if the procedures were not related to proteomics standards. The use of TFA (Trifluoroacetic acid) explained as Tallow Fatty Acid leave holes in the understanding which expertise in Proteomics the authors possess.

Author Response

Point 1: The introduction is well written even if the phenomenon of immunity induction by inactivated elicitors from fungi is not well documented by other examples/references from other species.

Response 1: We appreciate the reviewer’s important comment. In the introduction section, references [7-10] are the induction of potato, tomato seedlings and maize by pathogenic toxins. The inactivation of pathogenic toxins as an inducer is an innovation of this study, and no other plant reports have been reported.

Point 2: Results showed a gradual explanation of the techniques used and the graphs are quite illustrative of the control versus elicited response treatment. Not so evident results the reading of the Figure 2 and 3 which does not get too much explanation either in the results or in the discussion places, leaving the reader to a more visual understanding of what is going on without focusing on the key elements being up/down regulated.

Response 2: We appreciate the reviewer’s important comment. Some explanations have been added as belows:

Part of Figure 2: The results show that a significant enrichment of biological processes mainly affects the single-organism metabolic process (GO:0044710; 80 proteins; Pvalue_0.00000000141) and carbohydrate metabolic process (GO:0005975; 38 proteins; Pvalue_0.000000339). They all appear to activate the up-regulation. Carbohydrate and energy adjustment respond to plants’ responses to pathogen stress, providing the necessary dynamic support through the up-regulation of energy metabolism-related proteins to the resistance stress pathway.

Part of Figure 3: The metabolic pathways with the highest number of differentially expressed proteins are metabolic pathways (map01100; 53 proteins; Pvalue_0.000123), biosynthesis of secondary metabolites (map01110; 39 proteins; Pvalue_0.0000117), and phenylpropanoid biosynthesis (map00940; 13 proteins; Pvalue_0.0000346). The phenylpropanoid biosynthesis is related to plant defense reactions. Through a series of reactions, polyphenols such as lignin, flavonoids and flavonols, which can be synthesized can effectively prevent the invasion of various pathogens, and participate in the interaction process between plants and pathogens. Moreover, this signaling is associated with plant immune responses.

Point 3: The Supplementary figure 1 (S1) is showing the Gene Ontology accession up/down regulation sorted according P values but no info on other column legend meaning (i.e. PAS_Value PAS_Zscore max_zscore min_zscore, etc.).

Response 3: We appreciate the reviewer’s important comment. The column legend meanings have been added in the bottom of Supplementary Table S1 as belows:

maxLevel: maximal annotated level of this term in the GO graph (tree); Levels: a term can belong to different level in the GO graph. these levels are listed here, separated with ','; GO name: term name; GO ID: starting with GO, followed with a interge number, each GO term has a unique GO ID; P Value: calculated with Fish exact test with Hypergeometric algorithm; P value-adjusted: using 'Benjamini-Hochberg' method for multiple tests; PAS Value: the Pathway Activation Strength value, served as the activation profiles of the Signaling pathways based on the expression of individual genes; PAS-zscore: the standardize score of Pathway Activation Strength; max-zscore: the maximum value of the PAS-zscore; min-zscore: the minimum value of the PAS-zscore; Genes: list of involved genes in the query with this term. Fold change information is listed after the gene name separated with '|', if exists; Count: number of genes/proteins in the query that are involved in this term; Pop Hit: total number of consistent genes/proteins of this term in database; List Total: total number of differential proteins submitted; Background Genes: total number of known genes/proteins in a selected species.

Point 4: Furthermore a lot of abbreviation are not given as meaning (not even an abbreviation section exists): PPI = protein-protein interaction (used for establishing protein networks). Section 2.4 describe PPI analysis but leaves a lot of results from the Figure 4 without further interpretation leaving only the primary and secondary metabolite analysed in the discussion, meanwhile the antibiotics node-cluster is forgotten... 

Response 4: We appreciate the reviewer’s important comment. In section 2.4, PPI has been given the meaning as “In the PPI (protein-protein interaction) network map (used for establishing protein networks) (Figure 4)”.

    In addition, this section has been added as follows:

“In the PPI (protein-protein interaction) network map (used for establishing protein networks) (Figure 4), the most significantly enriched proteins, with high connectivity, are shown in red, and the significant enrichment of the metabolic pathway is shown in dark blue. The three metabolic pathways that were significantly enriched were Biosynthesis of secondary metabolites, Phenylpropanoid biosynthesis, and Metabolic pathways. There are 16 differential proteins directly related to the Biosynthesis of secondary metabolites pathway, nine of which are up-regulated; three differential proteins are directly linked to the Phenylpropanoid biosynthesis pathway, two of which are up-regulated; 22 differential proteins are linked to it in the Metabolic pathways pathway, 11 of which are up-regulated. A total of 11 differential proteins were up-regulated in this network map, and it was found that PH01000123G1100, PH01000159G0130, PH01000713G0340, PH01000761G0570 and PH01003309G0170 were associated with plant resistance expression, both of which showed significant up-regulation and at the same time affected multiple significantly enriched metabolic pathways.”.

Point 5: Other abbreviations to explain for the reader: PRM, Parallel reaction monitoring (a sort of Multiple Reaction Monitoring for peptides), etc.

Response 5: We appreciate the reviewer’s important comment. We have added the meaning of the abbreviations to the text. “PRM (Parallel reaction monitoring, a sort of Multiple Reaction Monitoring for peptides)” has been added in section 2.5.

Point 6: At the end the research output is restricted to 13 top differentially expressed protein candidates (Table 3) that discussed further for their possible involvement in the disease progression after exposure to the elicitor.

Still is very poor understood the response to AP-toxin, regarding its intrinsic mechanism of action, but now after this study maybe proteins involved in plant immunity could be sorted out from the KEGG databases sorted table (S2) and put into a contest, trying to understand their possible involvement.

Response 6: We appreciate the reviewer’s important comment. We think PRM combined with LC/MS quantitative analysis and TMT can explain this purpose. The reasons are as follows: 1) The PRM was used to verify the proteome. 2) From the perspective of influencing plant resistance, the related resistance proteins were screened according to the KEGG pathway and GO annotation, KEGG databases indicated that differentially expressed proteins are enriched from metabolic pathways, but we still need to verify these results. So we screened 14 candidate resistance proteins from KEGG database and GO annotation to verify using PRM, and the variables of proteins in PRM indicated the difference in response to AP-toxin, the 13 differentially expressed protein candidates still corresponded to the KEGG and GO databases. 3) The results of the screening were 14 candidate resistance proteins, 13 of which showed similar results to TMT. This result indicates that the TMT experiment is reliable. To facilitate a better understanding, some sentences have been added in Discussion section: “By PRM combined with LC/MS quantitative analysis to verify TMT results, 14 related proteins were screened from KEGG and GO annotation and functional classification based on the influence of plant resistance. The results showed that 13 candidate proteins showed similar trends to TMT, indicating TMT experiment. The data is reliable.”.

Point 7: Material and Methods seems to be well organized even if the procedures were not related to proteomics standards. The use of TFA (Trifluoroacetic acid) explained as Tallow Fatty Acid leave holes in the understanding which expertise in Proteomics the authors possess.

Response 7: We appreciate the reviewer’s important comment. This is our negligence. “Tallow Fatty Acid (TFA)” has been changed with “trifluoroacetic acid”. "TFA" should be trifluoroacetic acid,The TFA used in the experiment was mainly used for desalting the peptide after enzymatic hydrolysis of the sample, and TFA was not used in the LC-MS test.

Once again, thank you very much for your comments and suggestions.

Reviewer 3 Report

The manuscript “Differential Proteomics Based on TMT and PRM Reveal the Resistance Response of Bambusa pervariabilis × Dendrocalamopisis grandis Induced by AP-Toxin” presented by by Qianqian He and co-workers analyses the proteome of bamboo upon inoculation with the toxin produced by the induced resistance to Arthrinium phaeospermum. The work presented is innovative and interesting. However, in my opinion, the points below should be clarified to prior its publication in. In addition, proofing of the English is required as well as an overall improvement of the manuscript clarity (some examples of minor points are listed in minor comments; nonetheless they don’t cover all mistakes found in the present manuscript).

The introduction needs to give a better focus on the previous works were a similar strategy was used. Please clarify in the examples given on lines 57-65, are relative to the same plant, otherwise there are several reviews on the use of proteomics to study plant response to fungal pathogens that should be mentioned (e.g. Proteomics of Plant Pathogenic Fungi (2010); Proteomic Studies Revealing Enigma of Plant–Pathogen Interaction(2018) ).

In lines 88-90: the authors state that “...The TMT technique combined with LC-MS/MS were used to explore the differential expression of proteins and metabolic pathways induced by the toxin in B. pervariabilis × D. grandis. In lines 328-336 the authors describe the treatment to the samples that include the inoculation with the fungal pathogen. Please clarify, the proteomic analyses was performed on the samples inoculated with the fungi or not.  To discuss the role of the toxin in the increased resistance it would be useful to have the effect of the toxin before the fungal inoculation.

Line 84/85 – the taxonomical identification of the fungus should be up-dated and completed.

Lines (introduction) and lines (materials & methods) – the authors refer to the inactivation of the toxin but do not state the reason why this was done. The rationale behind the use of 40 ug/mL is not explained also.

The yellow in figure fig 2 is not easily seen; maybe a gray scale could work better

The figure 5 does not have enough resolution.  

Minor comments:

1)      In my opinion, there are too many abbreviations in the abstract, use onlythe essential ones.

2)      Line 72 – “... phenylalanin ammo-nialyase (PAL)...”

3)      Line 75/76 – please clarify: “Therefore, the phytoalexin theory was established, which laid the foundation for the study of potato-induced disease resistance”

4)      line 356 "Tallow Fatty Acid (TFA) solution"

Author Response

Point 1: proofing of the English is required as well as an overall improvement of the manuscript clarity (some examples of minor points are listed in minor comments; nonetheless they don’t cover all mistakes found in the present manuscript).

Response 1: We appreciate the reviewer’s important comment. We have re-proofed the English of the whole manuscript, in addition, we had requested “English Language Editing” in MDPI to polish this manuscript before the first submission, and now we have requested them to re-edit the revised manuscript.

Point 2: The introduction needs to give a better focus on the previous works were a similar strategy was used. Please clarify in the examples given on lines 57-65, are relative to the same plant, otherwise there are several reviews on the use of proteomics to study plant response to fungal pathogens that should be mentioned (e.g. Proteomics of Plant Pathogenic Fungi (2010); Proteomic Studies Revealing Enigma of Plant–Pathogen Interaction (2018)).

Response 2: We appreciate the reviewer’s important comment. The lines 57-65 have been changed with “González-Fernández et al. [5] reviewed the progress of proteomics in plant fungal pathogens research, which is an excellent tool that can give us a great deal of information about fungal pathogenicity by high-throughput studies. This approach has allowed the identification of new fungal virulence factors by characterizing signal transduction or biochemical pathways and studying the fungal life cycle and their life-style. Similarly, in the plant-fungus intricacies, proteomics provides rapid insight and is expected to be one of the imminent and integrative tools in biological research [6]. Rustagi et al. [6] summarized more than seventeen important plant-fungal pathogen studies that used proteomics by methods including Two-Dimensional Polyacrylamide Gel Electrophoresis (2D-PAGE), Fluorescent Two-Dimensional Difference Gel Electrophoresis (2D-DIGE), Isotope-Coded Affinity Tags (ICAT), iTRAQ, Multidimensional Protein Identification Technology (MudPIT), and Mass Spectrometry (MS).”. And the references have been changed with “Proteomics of Plant Pathogenic Fungi (2010); Proteomic Studies Revealing Enigma of Plant–Pathogen Interaction (2018)”.

Point 3: In lines 88-90: the authors state that “...The TMT technique combined with LC-MS/MS were used to explore the differential expression of proteins and metabolic pathways induced by the toxin in B. pervariabilis × D. grandis. In lines 328-336 the authors describe the treatment to the samples that include the inoculation with the fungal pathogen. Please clarify, the proteomic analyses was performed on the samples inoculated with the fungi or not.  To discuss the role of the toxin in the increased resistance it would be useful to have the effect of the toxin before the fungal inoculation.

Response 3: We appreciate the reviewer’s important comment. The variables in the test were inactivated purified pathogenic toxins as inducers, 15 plant spray inducers, 15 strains of sterile water, and 30 plants were inoculated with the same amount of pathogen suspension after spraying. In this way, plant changes are observed to determine whether the inducer is functional.

To clarify it, “The TMT technique combined with LC-MS/MS were used to explore the differential expression of proteins and metabolic pathways induced by the toxin and then inoculated by A. phaeospermum in B. pervariabilis × D. grandis.” in Introduction section and “Fifteen plants were randomly selected and sprayed on the upper eight shoots by the inducer (inactivated AP-toxin) in 4.1.” in Materials and Methods section have been added.

Point 4: Line 84/85 – the taxonomical identification of the fungus should be up-dated and completed.

Response 4: We appreciate the reviewer’s important comment. We have verified the taxonomical identification, and it has been changed as “The pathogen A. phaeospermum belongs to Fungi, Dikarya, Ascomycota, Pezizomycotina, Sordariomycetes, Xylariomycetidae, Xylariales, Apiosporaceae, Arthrinium” in Line 84/85.

Point 5: Lines (introduction) and lines (materials & methods) – the authors refer to the inactivation of the toxin but do not state the reason why this was done. The rationale behind the use of 40 ug/mL is not explained also.

Response 5: We appreciate the reviewer’s important comment. In the screening of the best inducer in the early stage, gradient concentrations were set for the inactivation temperature and pure toxin respectively, and the optimal inducer was determined by multiple comparisons. This result had been showed in our previous study in reference 17. To clarify it , we have added some illustrations in Introduction section, “Through the results of a previous study, the best inducer was the purified pathogen toxin inactivated at 60°C, and the concentration was 40 μg/mL [17].”.

Point 6: The yellow in figure fig 2 is not easily seen; maybe a gray scale could work better

The figure 5 does not have enough resolution.  

Response 6: We appreciate the reviewer’s important comment. The yellow in Figure 2 has been changed as green easily to see. And Figure 5 has been moved to supplementary material as Figure S1.

Minor comments:

Point 7: In my opinion, there are too many abbreviations in the abstract, use only the essential ones.

Response 7: We appreciate the reviewer’s important comment. Unnecessary abbreviations have been deleted in Abstract.

Point 8:  Line 72 – “... phenylalanin ammo-nialyase (PAL)...”

Response 8: We are very sorry for our negligence. “phenylalanin ammo-nialyase (PAL)” has been changed with “phenylalanine ammonia-lyase (PAL)”.

Point 9:  Line 75/76 – please clarify: “Therefore, the phytoalexin theory was established, which laid the foundation for the study of potato-induced disease resistance”

Response 9: We appreciate the reviewer’s important comment. This sentence has been changed with “this phytoalexin has provided the basis for the study of plant induced disease resistance.”.

Point 10: line 356 "Tallow Fatty Acid (TFA) solution"

Response 10: We are very sorry for our negligence. The abbreviations TFA "TFA" should be “trifluoroacetic acid”.

Once again, thank you very much for your comments and suggestions.

Round 2

Reviewer 2 Report

The revised paper looks much better now, even if this paper is clearly a proteomics paper, meaning that many of the tables and figures are not of immediate understanding for the readers of the "Metablolites" journal.